# A 2D Convection-Diffusion Model of Anodic Oxidation of Organic Compounds Mediated by Hydroxyl Radicals Using Porous Reactive Electrochemical Membrane

**DOI:** 10.3390/membranes10050102

**Published:** 2020-05-16

**Authors:** Ekaterina Skolotneva, Clement Trellu, Marc Cretin, Semyon Mareev

**Affiliations:** 1Physical Chemistry Department, Kuban State University, 149 Stavropolskaya str., 350040 Krasnodar, Russia; ek.skolotneva@gmail.com; 2Laboratoire Géomatériaux et Environnement (EA 4508), Université Gustave Eiffel, 77454 Marne la Vallée, France; clement.trellu@univ-eiffel.fr; 3Institut Europeen des Membranes, IEM-UMR 5635, ENSCM, CNRS, Univ Montpellier, 34095 Montpellier, France; marc.cretin@umontpellier.fr

**Keywords:** reactive electrochemical membrane, porous electrode, anodic oxidation, hydroxyl radicals

## Abstract

In recent years, electrochemical methods utilizing reactive electrochemical membranes (REM) have been considered as a promising technology for efficient degradation and mineralization of organic compounds in natural, industrial and municipal wastewaters. In this paper, we propose a two-dimensional (2D) convection-diffusion-reaction model concerning the transport and reaction of organic species with hydroxyl radicals generated at a TiO_x_ REM operated in flow-through mode. It allows the determination of unknown parameters of the system by treatment of experimental data and predicts the behavior of the electrolysis setup. There is a good agreement in the calculated and experimental degradation rate of a model pollutant at different permeate fluxes and current densities. The model also provides an understanding of the current density distribution over an electrically heterogeneous surface and its effect on the distribution profile of hydroxyl radicals and diluted species. It was shown that the percentage of the removal of paracetamol increases with decreasing the pore radius and/or increasing the porosity. The effect becomes more pronounced as the current density increases. The model highlights how convection, diffusion and reaction limitations have to be taken into consideration for understanding the effectiveness of the process.

## 1. Introduction

The removal of biorefractory emerging organic pollutants requires the implementation of a novel advanced treatment system for drinking water production and wastewater treatment. The use of membrane processes also requires an appropriate pretreatment in order to reduce membrane fouling [1,2,3]. In the last decades, anodic oxidation has been considered as a highly efficient electrochemical advanced oxidation process used in the treatment of natural and wastewaters containing organic pollutants [4,5]. The process involves the generation of hydroxyl radicals during the electrochemical decomposition of water and the consequent decomposition of organic compounds [4]. The hydroxyl radical is a highly reactive substance that allows the non-selective oxidation of most bio-degradable and bio-refractory organic pollutants [6]. Comparative studies show that among various methods, anodic oxidation allows achieving the highest mineralization of such compounds [5,7,8,9,10].

Despite all the advantages, the use of anodic oxidation in water purification processes is hampered by a number of drawbacks. First of all, hydroxyl radicals are formed only on the surface of the anode and have a short lifetime. Therefore, they are only present in a thin boundary layer (<1 μm) [11,12,13,14]. As a result, the oxidation of organic pollutants occurs only near the surface of the electrode. The process is limited by convective-diffusion delivery of contaminants from the solution to the reaction area. The most effective way to reduce mass transport limitations is to increase the electrochemically active surface area of the anodes by using porous electrodes, so-called reactive electrochemical membranes (REM), in flow-through configurations [15,16]. The effluent to treat is flowing through the porous anode. Several recent studies have focused on electrodes made from Magneli phases [17,18]. The results obtained over the past five years have shown that it is a very promising technology for electro-oxidation of organic pollutants in water treatment systems [19]. Such a material has a high oxygen evolution potential, which makes possible the generation of hydroxyl radicals, and thus, the oxidation of persistent organic compounds [20]. It has also been shown that porous anodes based on sub-stoichiometric titanium oxide exhibit suitable properties for efficient oxidation of organic pollutants [17,21].

Operating conditions drastically affect the efficiency of the electro-oxidation process. Electro-oxidation processes can be either operated under mass transport or current (reaction) limitation. In the case of porous electrodes, the flux density of the permeate and its residence time in the reaction zone are crucial parameters influencing mass transfer and reaction phenomena. An important role is also played by the distribution of the electric field strength in the pores of the electrode, the value of which is rather large, mainly on the side of the anode facing the cathode [16]. Nevertheless, it has been reported that the current density can be similar within the whole membrane depth when the process is operated under current limitation and without significant ohmic drop [18]. Another specificity of porous electrodes operated under current limitation has been reported: pollutants that are not degraded on the electrode surface can go deep into the anode, where the voltage is lower and not enough to generate hydroxyl radicals, but optimal for electropolymerization and further deposition of polymerized compounds [16,21,22], which might cause fouling and reduce process efficiency [21,23]. In [16], it was shown that a high current density significantly increased energy consumption, but allowed avoiding electro-polymerization due to an excessive amount of hydroxyl radicals and operation of the process under mass transport limitation. In recent studies, the influence of specific system parameters (flow rate, solution concentration and current intensity) has been experimentally studied as regards to both degree of purification achieved and energy consumption [19,24].

Mathematical modeling in the field of anodic oxidation for water treatment is also developing and has already made possible the estimation of the thickness of the reaction zone of anodic oxidation for dense electrodes [11,12,13], the development of transmission line models for estimation of the reactive region and REM fouling [16], the estimation of one-dimensional (1D) distribution of potential and current density within REM [18], the evolution of the concentration of organic compounds [18,25] or the prediction of permeate flux in tubular porous electrodes [26]. Convective-diffusion models that take into account hydroxyl radical mediated chemical reactions at the REM are absent, but their need has already been indicated in previous studies [21,23].

In this study, we present a theoretical analysis based on two-dimensional (2D) convection-diffusion-reaction model representing an anodic oxidation device with a porous electrode operated under galvanostatic condition. The model was calibrated based on experimental data previously obtained with a sub-stoichiometric titanium oxide REM [21]. Pore size, porosity and reaction rate between hydroxyl radicals and the model pollutant were taken from previous experimental studies. The model provides a better understanding of the distribution of current density over the heterogeneous porous surface. The model was then used for prediction of process effectiveness as a function of pore radius and porosity of the electrode.

## 2. Mathematical Model

### 2.1. Geometry of the System Under Study

The system under study is a cross-flow electrolyzer (Figure 1), which utilizes REM as an anode in inside-outside cross-flow filtration mode operated under galvanostatic conditions. The experimental setup was described in detail in a previous study [21]. The REM is a porous electrode (in our case, it is a tubular electrode with a defined inner, *r*_1_, and outer, *r*_2_, radii and length, L). A rod of radius *r*_3_ is used as a cathode and placed at the center of the REM. Anode and cathode form a tubular electrolyzer channel with a thickness (*r*_1_ − *r*_3_). A feed solution flows at a constant cross-flow rate, *U*. The permeate flux, expressed as the linear velocity *V*, is controlled by transmembrane pressure and depends on operating conditions of the experiment. The feed solution contains an organic compound with initial concentration *c*_0_, which is the target pollutant, and supporting electrolyte, which is assumed to do not participate in any chemical reaction and is used only to decrease the total resistance of the system. 

The real REM consists of a huge number of more or less heterogeneous pores, for which a three-dimensional (3D) model would be too complex to solve mathematically. The REM bulk might be considered as a homogeneous porous media characterized by only one structural parameter, for example, the inertial resistance factor [26]. In this case, the velocity and concentration distributions of the solution cannot be calculated in the individual pores, and it makes it more difficult to correctly define the effect of pore radius. Thus, in this work, we have applied other assumptions. The system under study consisted of a REM with the adjacent diffusion layer (DL) (Figure 1). The transition from a random to a hexagonal distribution of pores and conductive areas over the REM surface is shown in Figure 2a,b. The unit cell on the surface can be considered as consisting of a pore of radius *R*_1_ within a ring of anode material of external radius *R*_2_. The value of the curvature radius of the electrode surface was taken equal to the pore radius (Figure 2c) according to the image of the REM surface (Figure 2a). The DL thickness is *δ*. Then, the 3D unit cell was considered as a cylinder of radius *R*_2_, involving a pore and the surrounding part of the REM consisting of the solution with the DL of thickness *δ* (Figure 2c) calculated using the Leveque equation (Equation A15 in Appendix B). After the application of the cylindrical symmetry assumption, the system under study may be presented in two coordinates *r* and *z* (Figure 2d). The porosity of the modeled REM is calculated as *ε = (R*_1_*/R*_2_)^2^. The approach is applicable for different systems and allows us to describe the transport phenomena in systems with a complex structure [27,28].

### 2.2. Problem Formulation

According to previous theoretical investigations [11,12,13], the following simplifying assumptions are made:
The oxidation of organic compounds proceeds only via the assistance of hydroxyl radicals;The transport number of the organic compound is negligible comparing to the transport number of supporting electrolyte. Thus, only convection and diffusion fluxes are taken into account.Only the faradaic current is considered; the charging current is not taken into account;The temperature, activity coefficient and density gradients are ignored.The reaction of organic compounds by direct electron transfer on the REM surface is neglected.


Mathematical modeling of the mass transfer of chemical species in the vicinity of the REM includes the Fick’s law with the convective term (Equation (1)), the matter conservation (the continuity) law (Equation (2)), the Ohm’s law (Equation (3)), the charge conservation law (Equation (4)), the mass conservation law (Equation (6)) and the Navier-Stokes equation (Equation (5)), as follows:
(1)$${\vec{J}}_{i}=-D_{i}\nabla c_{i}+c_{i}\vec{V}$$
(2)$$\frac{\partial c_{i}}{\partial t}=-\mathrm{div}{\vec{J}}_{i}+v_{i}$$
(3)$$\vec{j}=-\kappa \nabla \varphi$$
(4)$$\mathrm{div}\vec{j}=0$$
(5)$$\frac{\partial \vec{V}}{\partial t}+\left(\vec{V}\;\nabla \right)\;\vec{V}=-\frac{1}{\rho_{0}}\nabla P+\nu \Delta \vec{V}$$
(6)$$\nabla \cdot \vec{V}=0$$
where *D*_i_, *c*_i_, and ${\vec{J}}_{i}$ are the diffusion coefficient, molar concentration and flux density of the *i*-th species, respectively; *v*_i_ is the chemical reaction, in which *i*-th specie is included; *φ* is the electric potential; *t* is the time; $\vec{V}$ is the linear velocity of the solution; *ρ*_0_ is the solution density (assumed to be constant); *P* is the pressure; *ν* is the kinematic viscosity; *κ* is the conductivity of the matter.

*P*, $\vec{V}$, *φ*, $\vec{j}$, ${\vec{J}}_{i}$ and *c*_i_ are functions of *t*, *z* and *r*. Equations (1)–(4) describe the concentration and potential fields, and Equations (5) and (6) describe the velocity field. These two groups of equations are coupled by the velocity in the Equation (1) and boundary conditions (Appendix B).

With regards to the modeling of (electro)-chemical reactions, the following reactions were considered. Hydroxyl radicals are generated from water discharge at the anode surface, according to the reaction [12]:
(7)$$H_{2}O\to HO_{\left(aq\right)}^{•}+H_{\left(aq\right)}^{+}+e^{-}$$


Free hydroxyl radicals can react with each other to form hydrogen peroxide:
(8)$$HO^{•}+HO^{•}\overset{k_{HO^{•}}}{\to }H_{2}O_{2}$$
where $k_{HO^{•}}$ is the rate constant of reaction (8).

In parallel to the reaction (8), hydroxyl radicals can also react with an organic compound (R) and its degradation by-products:
(9)$$R\overset{k_{R}}{\underset{HO•}{\to }}\mathrm{by-products}\overset{k_{i}}{\underset{HO•}{\to }}\mathrm{end-products}$$


Reaction kinetics are formulated assuming a second-order rate expression depending on the concentration of the hydroxyl radicals, the organic compound (*i* = 1) and its possible by-products (1 < *i* ≤ *n*):
(10)$$v_{HO^{•}}=k_{HO^{•}}c_{HO^{•}}^{2}+\sum_{i=1}^{n}k_{i}c_{HO^{•}}c_{i}$$
where *c*_HO•_ is the hydroxyl radical concentration and *c*_i_ and *k*_i_ are the concentration and the oxidation rate constant of the intermediate products in reaction (9).

Each by-product may have a distinct reaction coefficient *k*_i_ according to its reactivity with hydroxyl radicals [14]. For simplification, we used an average *k*_∑_ and integral *c*_∑_ (which is considered equal to the concentration of the organic compound *c*_R_) in our calculations. Thus, the second term of Equation (10) can be rewritten:
(11)$$\sum_{i=1}^{n}k_{i}c_{HO^{•}}c_{i}=\alpha k_{\Sigma }c_{HO^{•}}c_{R}$$
where *α* represents the number of hydroxyl radicals involved in the degradation of the initial organic compounds into end-products; *k*_∑_ is the average rate constant of reaction (9); *c*_R_ is the concentration of the organic compound. 

Thus, the rate of degradation of the initial organic compound oxidation reads:
(12)$$v_{R}=k_{R}c_{R}c_{HO^{•}}$$


In the stationary state condition, Equation (2) with Equations (10)–(12) becomes as follows:
(13)$$\mathrm{div}{\vec{J}}_{HO^{\cdot }}=k_{HO^{•}}c_{HO^{•}}^{2}+\alpha k_{\Sigma }c_{R}c_{HO^{•}}$$
(14)$$\mathrm{div}{\vec{J}}_{R}=k_{R}c_{R}c_{HO^{•}}$$
where subindexes HO^•^ and *R* are related to the hydroxyl radical and organic compound, respectively.

The boundary conditions are presented in Appendix B.

## 3. Results and Discussions

Equation system (1)–(6) under the boundary conditions (Appendix B) is solved numerically using Comsol Multiphysics 5.5 software package.

### 3.1. Degrdation of Paracetamol and its by-Products

In the calculations, the oxidation of paracetamol (PCT) was considered. Brillas et al. [29,30] proposed several possible ways for PCT oxidation by hydroxyl radicals. Its well-known degradation by-product 1,4-benzoquinone (Figure 3) is strongly refractory to direct electron transfer. But some end-products such as carboxylic acids have a lower oxidation rate by HO^•^ and faster mineralization can be achieved by DET [31]. Therefore, it was calculated that 28 hydroxyl radicals participate in the oxidation of PCT into oxalic and oxamic acids. Thereby, we used *α* = 28 in all the calculations presented in this study.

### 3.2. Treatment of Experimental Data

The experimental data are taken from [21]. The parameters of the experimental setup were as follows: the REM was Magnelli phase anode with *r*_1_ = 3 mm, *r*_2_ = 5 mm, *L* = 9 cm, the stainless steel rod was used as cathode with *r*_3_ = 1.5 mm, the cross-flow rate was *U* = 0.88 m·s^−1^, the permeate flux rate (*V*) varied in the range 0.44–5.4 × 10^−4^ m·s^−1^, the concentration of PCT was *c*_R_^0^ = 0.19 mM, concentration of the supporting electrolyte (sodium sulfate) was 50 mM, current density (*j*) was in the range 60–300 A·m^−2^. The REM has a monomodal pore size distribution ranging between 0.8 and 1.9 μm. The median pore size *R*_1_ = 0.7 µm was considered for the model. The relative porous fraction considered at the REM/liquid interface was *ε* = 0.2 (value obtained from the analysis of SEM images, Appendix A).

The value of *δ* = 30 μm is obtained from the Lévêque approximate solution (A15) according to hydrodynamic conditions and geometric parameters of the device used in the experiments. The rate constant of PCT oxidation reaction *k*_R_ was evaluated in numerous articles [21,32,33,34], and it differs in the range between 2 × 10^6^ and 1.4 × 10^7^ m^3^mol^−1^·s^−1^. We used an average value of *k*_R_ = 1 × 10^7^ m^3^mol^−1^·s^−1^. The value of ***k_Σ_*** was considered as a fitting parameter and was evaluated by comparing experimental and theoretical results (Figure 4).

Experimental and theoretical curves of (PR) of PCT as a function of total organic carbon (TOC) flux through the REM are shown in Figure 4. During the calculations, we varied the permeate flux (velocity) of the solution while the concentration remained constant. The graphs show that both in the experiment and theory, the PR strongly depends on permeate flux, which both increases the amount of organic compounds to oxidize and decreases the residence time of organic compounds in the reaction zone. As expected, PR decreases when increasing the permeate flux. Interestingly, the PR starts to decrease below 99% only from a threshold value that strongly depends on the current density. At low permeate flux, the process is operated under mass transport limitation, meaning that the amount of organic compounds to oxidize is limited by the transport of these organic compounds to the reaction zone. At high permeate flux, the process can switch from mass transport to reaction limitation, meaning that the oxidation rate is limited by the amount of hydroxyl radicals generated in the reaction zone. This latter limitation results in the decrease of the PR below 99%. As the formation of hydroxyl radicals is promoted by the increase in current density, it is consistent to observe that the reaction limitation occurs at higher permeate flux for high current density. A complete explanation of the results should also take into consideration both diffusive and convective mass transport of organic compounds. If the velocity of the solution is high, the convective flux exceeds the diffusion, and some organic molecules may not have the time to reach the reactive zone at the REM surface. This phenomenon will be further discussed below.

### 3.3. Concentration and Current Density Distribution

The current lines in the system are distributed non-uniformly over the electrode surface (Figure 5a). They condense at the upper side and become sparser in the bulk of the pore. The heterogeneous distribution of the current density leads to strong increases of local current densities on the REM surface. As can be seen, the model developed in this study predicts that the active area of the REM surface is located at the entrance of the pore (Figure 5a) [16]. Thereby, the local flux of hydroxyl radicals, which depends on the current density (Equation (15)), also enhances the reactive area at the upper side of the electrode/solution interface.

Previous studies have highlighted the competition between reactions (8) (hydroxyl radical dimerization) and (9) (hydroxyl radical reaction with organic compounds) and the influence of organic compounds on the distribution profile of hydroxyl radicals: the higher the concentration of organic compounds, the lower the reactive zone thickness is [10,11,12]. The analytical Equation (15), which takes into account both reactions, was considered for taking into account the influence of the concentration of the organic compound (*c*_R_) on the hydroxyl radical concentration distribution (Appendix C):
(15)$$c_{HO^{•}}=\frac{4\alpha k_{\Sigma }c_{R}c_{HO^{•}}^{s}}{{\left[\sqrt{\frac{2}{3}k_{HO^{•}}c_{HO^{•}}^{s}+zk_{\Sigma }c_{R}}\left(1-A\right)+\sqrt{zk_{\Sigma }c_{R}}\left(1+A\right)\right]}^{2}}A;\;A=\exp\left(-x\sqrt{\frac{\alpha k_{\Sigma }c_{R}}{D_{HO^{•}}}}\right)$$
where $c_{HO^{•}}^{s}$ is the concentration of HO^•^ on the anode surface.

The concentration profiles of hydroxyl radicals and organic compounds are shown in Figure 5. The concentration of hydroxyl radicals exponentially depends on the distance from the electrode surface (Figure 5d). The distribution is typical for the anodic oxidation systems [11,12,13]. Previous theoretical studies demonstrated that the thickness of the reaction zone is a function of current density, organic compound concentration, diffusion coefficient, reaction rate constant and number of hydroxyl radicals involved in the oxidation of one molecule. A good agreement was obtained between our results and previous studies: the reaction zone thickness varied between 1 nm (in the presence of 1 M of organic compound) and 1 μm in the absence of organics [12]. In the experimental conditions detailed above, results from the model indicates that hydroxyl radicals are only present in a reaction zone thickness lower than 300 nm from the electrode surface. As this thickness is lower than pore radius or DL thickness, it means that diffusion of the organic compounds might be a crucial phenomenon. Therefore, parameters such as convection rate, pore size and porosity of the electrode have to be considered in order to ensure that organic pollutants have sufficient time for diffusing to the reaction zone.

### 3.4. Effect of REM Pore Radius

For different current densities (*j* = 60, 150 and 300 A·m^2^), we have predicted from the model the evolution of the PR of PCT at fixed porosity (*ε* = 0.2) and various length of pore radius *R*_1_. As can be seen in Figure 6, the PR of PCT decreases with increasing *R*_1_ at fixed TOC flux. The effect is due to the diffusion limitation of organic compound delivery to the surface of the electrode, where the reactive zone is located. The characteristic time for PCT diffusion to the electrode surface decreases with low pore size because the distance from the center of the pore to the electrode surface is shorter. Thus, at a fixed permeate flux (convection), the probability of a molecule to reach the reaction zone increases with decreasing *R*_1_.

This limitation from the diffusion also depends on the permeate flux. The limitation from the diffusion actually starts when the convection becomes too high compared to the times required for diffusion. At low permeate flux, organic compounds will have sufficient time to diffuse to the electrode surface, whatever the length of pore radius (in the range studied). Thus, at TOC flux below 5 g·m^−2^·h^−1^, similar PR of PCT was predicted for all values of pore radius. Interestingly, at low current density (60 A·m^−2^), the model did not predict any significant effect of pore radius in the range 0.5–1.5 µm. It means that the diffusion was not the limiting phenomenon in this case. Instead, limitation from the reaction was the predominant phenomenon controlling the PR of PCT. At higher current density, limitation from the reaction is reached at higher permeate flux, thus making possible the limitation from the diffusion to occur and to affect the PR of PCT. For example, with a pore radius of 5 µm, similar trends were obtained whatever the current density, meaning that the limitation from the diffusion was the predominant phenomenon in this different case.

These results predicted from the model highlight that convection, diffusion and reaction phenomena have to be taken into consideration in order to optimize crucial parameters of the process such as pore size, permeate flux and current density.

### 3.5. Effect of REM Porosity

Similar to the previous case, we fixed one of the geometric parameters, namely the pore radius: *R*_1_ = 0.7 μm. We analyzed the effect of *ε* on the PR of PCT at different current density (Figure 7). Results predicted from the models indicate that the effectiveness of the process should be improved when increasing the porosity. For a constant pore radius, increasing the porosity means that the number of pores over a same surface is increasing. For the same permeate flux, it involves that the water velocity (convection) in each pore is lower, and thus, pollutants have more time to diffuse to the electrode surface. Therefore, increasing the porosity allows the limitation from the diffusion to occur only at higher permeate flux. As observed for pore radius, the effect of porosity is not significant for the lowest current density (60 A·m^−2^) because in this case the PR of PCT is almost only controlled by the limitation from the reaction that occurs at low permeate flux.

Overall, these theoretical results highlight how current density, pore radius and porosity influence the process effectiveness of the REM by taking into consideration convection, diffusion and reaction phenomena. Such models might help to the design of novel efficient REM minimizing mass transport limitations.

Nevertheless, it might be also important to take into consideration an additional effect, which is the influence of the porosity on the distribution of the current density (Figure 8). When the porosity increases, the surface of the electrode material at z = 0 decreases. As the active area of the REM is mainly located at the entrance of the pore, it leads to an increase of local current densities (Figure 8). This effect might have adverse effects on the lifetime of the electrode since high local current density might promote passivation and corrosion of the electrode.

## 4. Conclusions

In the paper, we proposed a 2D stationary model of transport of diluted species in the electrolysis system with a REM. The model is based on the Fick-Navier-Stokes equations and takes into account the local geometrical and hydrodynamic properties of the system as well as chemical reactions related to the oxidation of organic compounds by hydroxyl radicals. Porosity and pore radius of the REM were considered as two crucial parameters.

We highlighted theoretically the competitive mechanisms relative to the delivery of an organic compound to the reactive zone, where the oxidation reaction occurs. Convection, diffusion and reaction limitations have to be taken into consideration in order to explain the results. The calculated dependencies of the degradation rate of a model organic compound on TOC flux are in good agreement with experimental data.

Porosity and pore radius of the REM significantly affect the removal of the target organic compound: the degradation rate decreases with increasing pore radius or decreasing porosity. These adverse effects are due to the diffusion time of organic compounds to the pore surface that becomes limiting compared to the characteristic time of convection. These theoretical predictions might be useful for the conception of novel REM minimizing mass transport limitations.

## Figures and Tables

**Figure 1 membranes-10-00102-f001:**
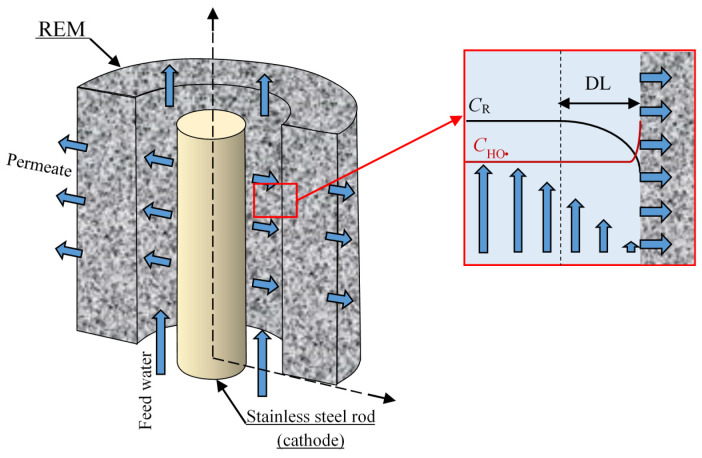
Schematic representation of the system under study. *C_R_* represents the concentration of an organic compound. The red square shows the area taken into consideration for the model: the electrode and the diffusion layer (DL).

**Figure 2 membranes-10-00102-f002:**
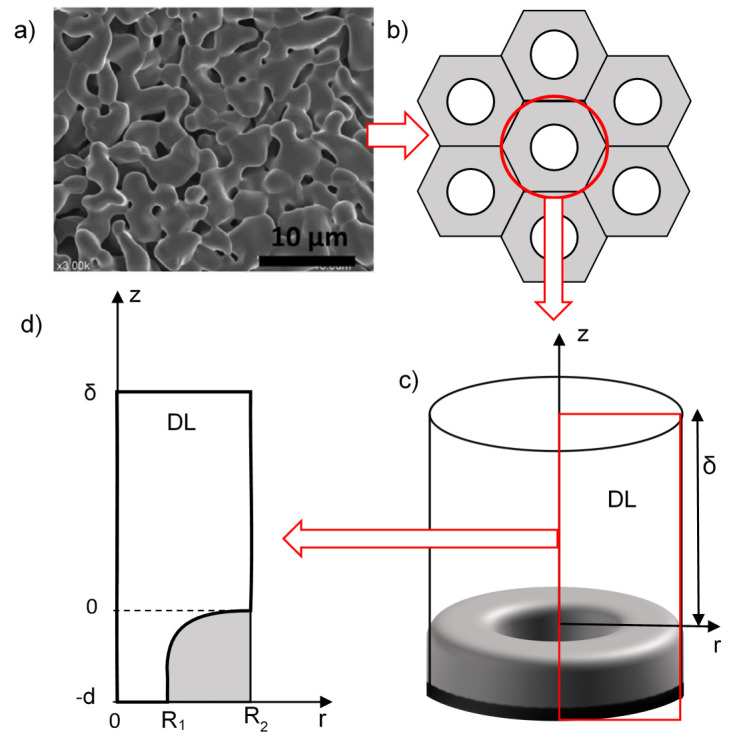
The schematic representation of the transition from a complex surface structure of the reactive electrochemical membranes (REM) (**a**) to the simplified uniformly distributed pattern (**b**) and three-dimensional (3D) model representation with axial symmetry (**c**) and two-dimensional (2D) unit cell (**d**).

**Figure 3 membranes-10-00102-f003:**
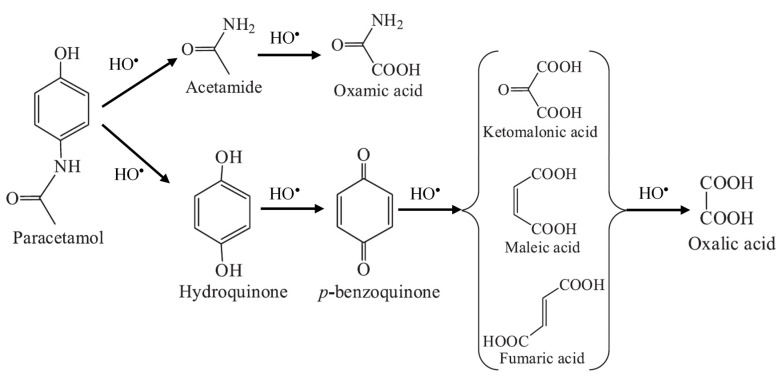
Proposed reaction sequence for paracetamol degradation in acidic aqueous medium by hydroxyl radicals oxidation processes [30]. In our study, we have considered the hydroxyl radical mediated oxidation of paracetamol into oxalic and oxamic acids.

**Figure 4 membranes-10-00102-f004:**
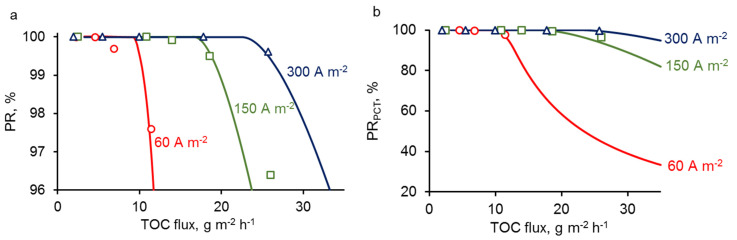
Experimental (dots) and theoretical (lines) dependencies of percentage removal of PCT (PR_PCT_) on total organic carbon (TOC) flux at different current densities (mentioned in the figure). All other parameters are presented in Table 1. (**a**) is a zoomed version of (**b**).

**Figure 5 membranes-10-00102-f005:**
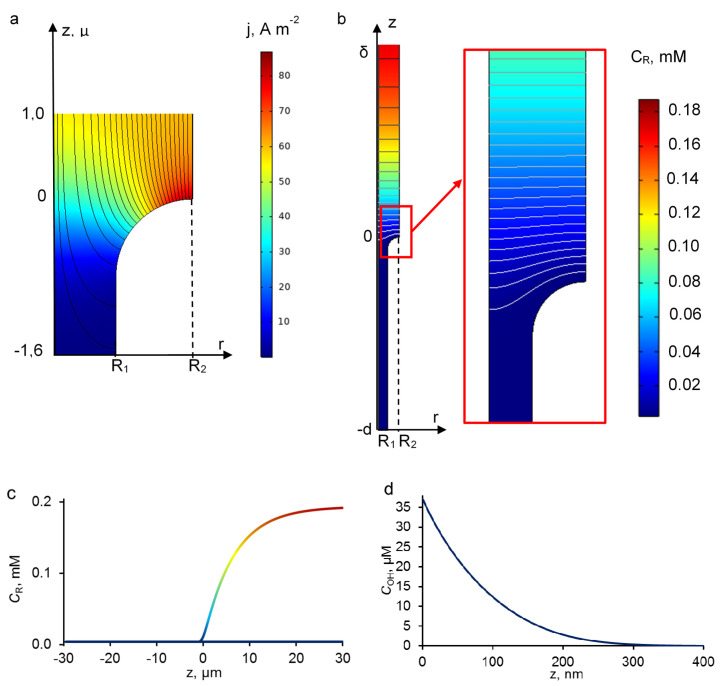
Calculated current density distribution and current lines near the entrance of the pore (**a**). Theoretical concentration distribution of PCT in the system (**b**), at the symmetry axis at *r* = 0 (**c**). Theoretical concentration distribution of HO^•^ radicals at *r* = *R*_2_ (**d**). The calculations were performed at *j* = 60 A·m^−2^; *V* = 0.0001 m/s. All other parameters are presented in Table 1.

**Figure 6 membranes-10-00102-f006:**
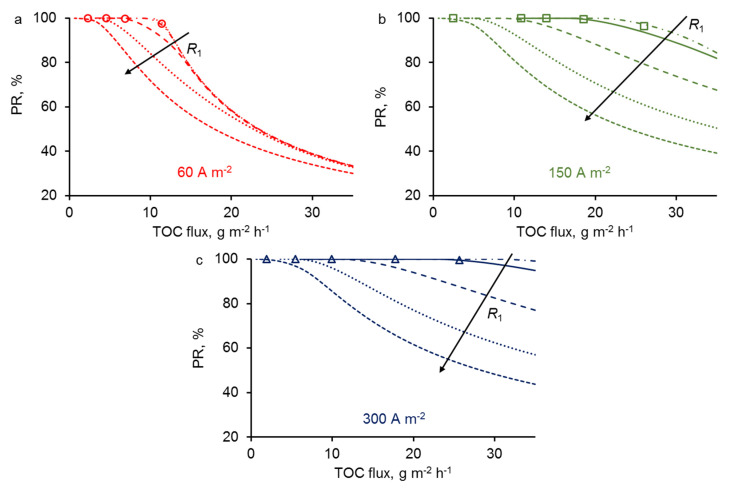
The influence of pore radii on the PR of the PCT at fixed porous fraction ε = 0.2 and *j* = 60 (**a**), 150 (**b**) and 300 (**c**) A·m^−2^. The current density is mentioned in each figure. The arrow represents an increment of pore radius: *R*_1_ = 0.5, 0.7, 1.5, 3 and 5 μm. All other parameters are presented in Table 1.

**Figure 7 membranes-10-00102-f007:**
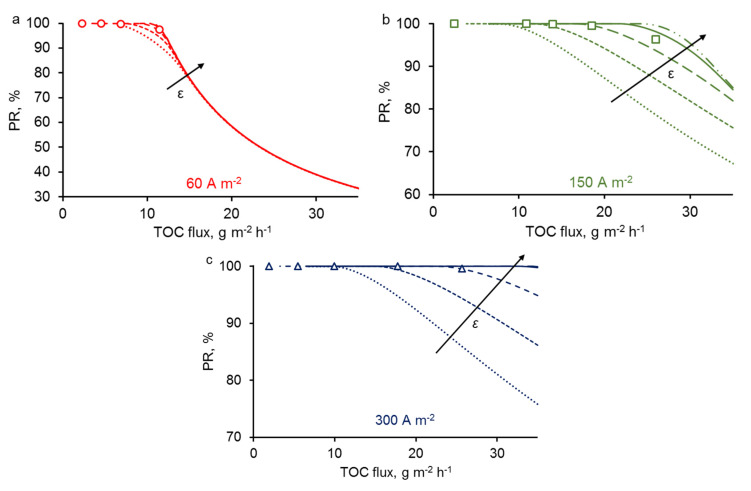
The influence of porosity on the PR at fixed pore size *R*_1_ = 0.7 μm and *j* = 60 (**a**), 150 (**b**) and 300 (**c**) A·m^−2^. The arrow represents an increment of porosity: *ε* = 0.05, 0.1, 0.2, 0.4, 0.6. All other parameters are presented in Table 1.

**Figure 8 membranes-10-00102-f008:**
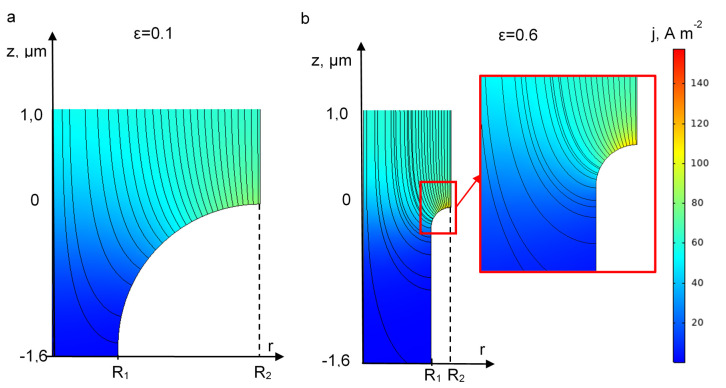
The influence of the REM porosity (mentioned in the pictures) on the local current density distributions at *j* = 60 A·m^−2^ and *ε* = 0.1 (**a**) and 0.6 (**b**). All other parameters are presented in Table 1.

**Table 1 membranes-10-00102-t001:** Parameters of the system used in the simulations. PCT: paracetamol.

*R*_1_, μm	*ε*	*k*_R_, m^3^mol^−1^·s^−1^	*k_Σ_*, m^3^mol^−1^·s^−1^	*k*_HO__•_, m^3^mol^−1^·s^−1^	*c*_R_^0^, mM	*D*_PCT_, m^2^/s	*D*_HO•_, m^2^/s	*α*	*δ*, μm
0.7 [21]	0.2 Suppl. M.	1 × 10^7^ [21,32,33,34]	6.5 × 10^6^ fitting parameter	5.5 × 10^6^ [35]	0.19 [21]	0.65 × 10^−9^ [36]	2.2 × 10^−9^ [12]	28	30 Equation (A15)

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
