# Peer review of "A 2D Convection-Diffusion Model of Anodic Oxidation of Organic Compounds Mediated by Hydroxyl Radicals Using Porous Reactive Electrochemical Membrane"

_membranes, 2020, doi:10.3390/membranes10050102_

Round 1

Reviewer 1 Report

Manuscript ID: membranes-801542

Title: A 2D convection-diffusion model of anodic oxidation of organic compounds using porous reactive electrochemical membrane

Reviewer comments _ 01 05 2020

The text is clearly written and is generally easy to follow. However, it contains various grammatical errors. There are also some inconsistencies in markup, fontsize etc.

It would be helpful if the authors can define objectives that their model should help to achieve in the introduction. The purpose and potential application of the model provide a framework to evaluate, for example, if assumptions can be justified, or if the experimental data is described well enough. This might also help with providing the reader with a clear message, which is somewhat lacking from the abstract/conclusion sections.

The modeling is based on many well established equations (e.g. Fick’s, continuity) and boundary conditions (e.g. no-slip). For readability, it is suggested to provide these in a table or appendix, while in the main text mainly discussing/highlighting equations that are specific for this situation / geometry or that are otherwise unusual.

line 197: “The DL thickness must be known to solve the problem... “  It seems like the set of equations presented, would be perfectly capable of solving the problem, if the boundary condition is set for the inlet of the system, rather than for the interface at the boundary layer. The latter approach seems more suitable for a 1D approximation, rather than 2D as is the case here. It would be helpful if the authors can elaborate on this topic, to justify their chosen approach.

The derivation of the equations for the reaction kinetics in eqns 8-12 etc. contains several large assumptions, which should be explicitly mentioned and if possible justified. For example:

  • Every compound has a distinct reaction coefficient ki and reaction order. Generally ∑ ki ≠ n*k. The more easily degradable compounds disappear first, whereas the more stable compounds persist. So if some average k and lumped c are used, it should be understood that the value of k may depend on the progression of the reactions.
  • The oxidation of one compound generally forms another compound. Thus, the total concentration of organics (∑ ci) does not necessarily decrease with the consumption of OH*

Figure 4: The experimental data describes the percentage removal of a specific compound (PCT), whereas the model describes a lump-sum of compounds. These two properties cannot be directly compared.

Figure 4: The data does not contain many distinct points that are not on the line PR≈100%. For the points that significantly less than 100%, the match between the model and the data does not appear to be that great at first glance. However, the evaluation of the match between model and experiment depends on the number of fit-parameters (should be explicitly mentioned here) and the purpose of the model.           

table 233: please indicate how these constants were obtained. Where any of these fit-parameters?

Author Response

Authors' reply

Dear Reviewer,

Thank you for your suggestions. We agree with your remarks and made all the necessary corrections (they are marked in yellow during the article). Some of them are presented in detail below.

Comment

The modeling is based on many well established equations (e.g. Fick’s, continuity) and boundary conditions (e.g. no-slip). For readability, it is suggested to provide these in a table or appendix, while in the main text mainly discussing/highlighting equations that are specific for this situation / geometry or that are otherwise unusual. 

Answer

Thank you for the comment. Now the boundary conditions are provided in Appendix A1. We believe that the main equation system should be in the main text.

Comment

line 197: “The DL thickness must be known to solve the problem... “  It seems like the set of equations presented, would be perfectly capable of solving the problem, if the boundary condition is set for the inlet of the system, rather than for the interface at the boundary layer. The latter approach seems more suitable for a 1D approximation, rather than 2D as is the case here. It would be helpful if the authors can elaborate on this topic, to justify their chosen approach.

Answer

We have added the following text into the model formulation section.

The real REM consists of a huge number of more or less heterogeneous pores, for which a 3D model would be too complex to solve mathematically. The REM bulk might be considered as a homogeneous porous media characterized by only one structural parameter, for example, the inertial resistance factor [25]. In this case, the velocity and concentration distributions of the solution cannot be calculated in the individual pores, and it makes more difficult to correctly define the effect of pore radius. Thus, in this work, we have applied other assumptions. The system under study consists of a REM with the adjacent diffusion layer (DL) (Figure 1).”

In this case our model may be considered as 1D extended to the 2D case to take into account the influence of the pore radius.

Comment

The derivation of the equations for the reaction kinetics in eqns 8-12 etc. contains several large assumptions, which should be explicitly mentioned and if possible justified. For example:

  • Every compound has a distinct reaction coefficient ki and reaction order. Generally ∑ ki ≠ n*k. The more easily degradable compounds disappear first, whereas the more stable compounds persist. So if some average k and lumped c are used, it should be understood that the value of k may depend on the progression of the reactions.

Answer

We have added the text, which is close to that you have proposed

“Each by-product may have a distinct reaction coefficient ki according to its reactivity with hydroxyl radicals [13]. For simplification, we used an average k and integral c (which is considered equal to the concentration of the organic compound cR) in our calculations. Thus, the second term of Eq. can be rewritten:”

Comment

  • The oxidation of one compound generally forms another compound. Thus, the total concentration of organics (∑ ci) does not necessarily decrease with the consumption of OH* Figure 4: The experimental data describes the percentage removal of a specific compound (PCT), whereas the model describes a lump-sum of compounds. These two properties cannot be directly compared.

Answer

Thank you for the comment. Yes, you are right, in the model, the consumption of OH* depends on a lump-sum of organic compounds, which is assumed to be equal to the PCT concentration in Eq. 13. The approach is common in the literature [10,11,13,14]. Nevertheless, the consumption of hydroxyl radicals is several times faster than the consumption of PCT due to taking into account the number of HO* radicals required for the oxidation of intermediate products. And if we will take into account the increase in the lump-sum of organic compounds during the oxidation of PCT, which is very interesting for future investigation, then, perhaps, we could obtain a similar situation with PCT concentration. The comparison of the approaches could be done in the next work. We have added the discussion in sections 3 and 4.

Comments

Figure 4: The data does not contain many distinct points that are not on the line PR≈100%. For the points that significantly less than 100%, the match between the model and the data does not appear to be that great at first glance. However, the evaluation of the match between model and experiment depends on the number of fit-parameters (should be explicitly mentioned here) and the purpose of the model.            

table 233: please indicate how these constants were obtained. Where any of these fit-parameters?

Answer

We have included the information about all the parameters in the table and in the description of all the Figures. Also, the supplemental materials were added.

Reviewer 2 Report

       This paper mainly builds a two-dimensional steady-state model about reactive electrochemical membrane and tries to solve such problems as the transmission of organic compound to reaction zone and its removal rate. My comments are as following:

  1. The size of words behind equations 17 to 20 is not consistent with that ahead of the equations.
  2. The serial numbers of line 223 and that behind line 253 are wrong.
  3. Just cite the most important and relative references. Do not cite many papers at a time. Avoid lumping references as in Ref. [1-5]and all other.
  4. The abstract and conclusion are not well-written thus should be polished. Try to make the conclusion more sharp and leave the take home message.
  5. Explanation of the past researches is unclear in the text (Details are unclear). Past researches in this field and their consequences should be fully explained. Additional explanations should be included to clarify the results of relevant past researches in the Introduction section.
  6. The experimental data is from one study. Is it representative? Please explain why you choose this study.

Author Response

Authors' reply

Dear Reviewer,

Thank you for the comments. We find your suggestions very fruitful and made all the necessary corrections (you can find them marked by the yellow color in the text).

According to the experimental data: the results were obtained by Clement Trellu (co-author of the work). The goal of the experiments was to achieve the maximum removal of the PCT and found the optimum TOC flux and current density. Low percentage removal was not considered. Nevertheless, your suggestions were added to the discussion section and conclusions.

Round 2

Reviewer 1 Report

The authors satisfactorily addressed the reviewer comments and revised the manuscript.
The revised manuscript is now considered suitable for publication.
Congratulations.